# Neural Rankers for Code Generation via Inter-Cluster Modeling

## Abstract

Code Large Language Models (CodeLLMs) have ushered in a new era of code generation advancements. However, selecting the best solutions from all possible CodeLLM solutions remains a challenge. Previous methods frequently overlooked the intricate functional similarities and interactions between clusters, resulting in suboptimal results. In this work, we introduce *SRank*, a novel reranking strategy for selecting the best solution from code generation that focuses on modeling inter-cluster relationship. By quantifying the functional overlap between clusters, our approach provides a better ranking strategy of code solutions. Empirical results show that our method achieves remarkable results on pass@1 score. For instance, on the Human-Eval benchmark, we achieve 69.66% in pass@1 with Codex002, 75.31% for WizardCoder, 53.99% for StarCoder and 60.55% for Code-Gen, which surpass the state-of-the-arts solution ranking methods, such as CodeT and Coder-Reviewer on the same CodeLLM with significant margin ($\approx 6.1\%$ improvement on average). Comparing to the random sampling method, we can achieve an average improvement of $\approx 23.07\%$ on Human-Eval and 17.64% on MBPP. Even in scenarios with limited test inputs, our approach demonstrates robustness and superiority, marking a new benchmark in code generation reranking.

## 1 Introduction

Recent advancements in language models tailored for code, known as Code Large Language Models (CodeLLMs) (Luo et al., 2023; Wang et al., 2023; Nijkamp et al., 2023b; Rozière et al., 2023), have garnered significant interest, particularly due to the expansion of large-scale language models and the volume of pre-training data (Kaplan et al., 2020; Zhao et al., 2023). A primary utility of CodeLLMs is their capacity to generate code given to natural language descriptions written by humans (Chen et al., 2021; Fried et al., 2023; Chowdhery et al., 2022; Nijkamp et al., 2023b). However, prior studies (Holtzman et al., 2020; Austin et al., 2021) have highlighted that the sequences generated by these models can be prone to errors, especially when likelihood-based decoding techniques like greedy search and beam search are employed. Alternatively, sampling-based decoding techniques (Fan et al., 2018; Holtzman et al., 2020) extract multiple solutions from the model's multinomial distribution. This method generates a wide range of code solutions, many of which are correct (Austin et al., 2021; Ni et al., 2023). As a result, there is a growing interest in developing reranking strategies for code generation (Li et al., 2022; Inala et al., 2022; Chen et al., 2023; Zhang et al., 2023; Ni et al., 2023), with the goal of sorting through an abundance of sampled solutions to identify high-quality and accurate ones.

The goal of reranking is to organize the set of candidate programs so that accurate programs are prioritized. Li et al. (2022), Chen et al. (2023), and Ni et al. (2023) have clustered code solutions based on their functionality, then used cluster-specific data to determine ranking scores. Given that language models frequently produce code solutions that differ syntactically but are semantically analogous, functional clustering narrows the candidate pool. The emphasis then shifts from ranking individual solutions to ranking clusters themselves. Previous ranking strategies, such as AlphaCode (Li et al., 2022), CodeT (Chen et al., 2023), and Coder-Reviewer (Zhang et al., 2023), provide approaches to clustering and reranking code solutions. While AlphaCode (Li et al., 2022) focuses on identical outputs from model-generated test inputs, CodeT (Chen et al., 2023) focuses on solutions that pass model-generated test cases. The Coder-Reviewer approach (Zhang et al., 2023), inspired by collaborative software development, uses a dual-model system to cross-check generated

programs against language instructions. However, by treating clusters in isolation, they fail to model potentially informative functional similarities and interactions across clusters.

To address this limitation, we propose **SRank**, a novel reranking approach emphasizing *modeling inter-cluster* relationships. Specifically, we introduce a new metric called *functional overlap* to quantify the similarity between clusters based on their execution outputs. This allows for identifying the most representative cluster that exhibits maximal overlap with all other clusters. As inconsistencies often indicate incorrect functionality, the cluster interacting most comprehensively likely represents the optimal solution. By incorporating these inter-cluster relationships into the ranking pipeline, we can better identify the most promising solutions. Through extensive evaluation, we demonstrate that modeling inter-cluster relationships and functional overlap provides significant and consistent improvements over prior state-of-the-art solution ranking methods (Li et al., 2022; Chen et al., 2023; Zhang et al., 2023) on a wide range of state-of-the-arts CodeLLMs, including Codex, WizardCoder, StarCoder and CodeGen. For instance, on the HumanEval benchmark, our method achieved a pass@1 score of 75.31% with WizardCoder34B, outperforming the Coder-Reviewer's score of 66.9%. Similarly, on the MBPP benchmark, our method improved the pass@1 score for WizardCoder from 50.3% with CoderReviewer to 51.03% with our approach. Similar improvements are applied for other CodeLLMs, including StarCoder, CodeGen and Codex002. If we compare SRankwith a simple random sampling method to get the solutions, we observe massive improvements across the models, with average improvements of 23.07% and 17.64% for Human-Eval and MBPP, respectively. Our evaluation is more *comprehensive* because we include many SOTA CodeLLMs of varying sizes, whereas CodeT and Coder-Reviewer did not. This provides compelling evidence of our approach's robustness across a wide range of models.

We also conducted an extension ablation study to demonstrate some of our advantages, such as our approach's remarkable robustness even with limited test cases. In summary, by moving from isolated clusters to interacting clusters with quantified functional overlap, our novel reranking strategy aims to address the limitations of prior ranking techniques for code generation. To summarize our contributions, they are as follows:

- We introduce a novel reranking strategy for CodeLLMs that emphasizes the inter-cluster relationships and leverages the functional overlap between them, providing a more robust and accurate approach to pick the best solutions.
- Through extensive and comprehensive evaluations, we demonstrate that our approach consistently outperforms existing state-of-the-art methods in code generation. For instance, our method achieved superior results on both the HumanEval and MBPP-S benchmarks across various CodeLLMs.
- We perform extensive ablation study to evaluate the robustness of our method, highlighting its effectiveness even with a limited number of test inputs and its ability to capture intricate interactions between clusters, setting it apart from previously isolated ranking techniques.

## 2 BACKGROUND & MOTIVATION

### 2.1 CODE GENERATION

Code generation is the process of solving programming problems by generating code solutions based on a given context $c$. The context includes a natural language description and a code snippet that contains statements such as imports and a function signature. In addition to this, a predefined set of test cases, denoted as $T$, is provided to evaluate the correctness of the generated code solutions. Using $c$ as the input on CodeLLM, we can a set of solutions $\mathbf{P} = \{p_1, p_2, ..., p_N\}$, where N is hyperparameter that define number of return sequences from the execution of CodeLLM. A solution $p$ is considered valid if it successfully passes the predefined set of test cases $\mathbf{T}$.

### 2.2 SOLUTION CLUSTERING AND RERANKING

The reranking task aims to prioritize correct programs in the candidate list $\mathbf{P}$. Previously, solutions are clustered by functionality, simplifying the task due to the language models' tendency to produce syntactically varied but semantically similar solutions. Thus, the focus shifts from ranking individual solutions to ranking these functional clusters.

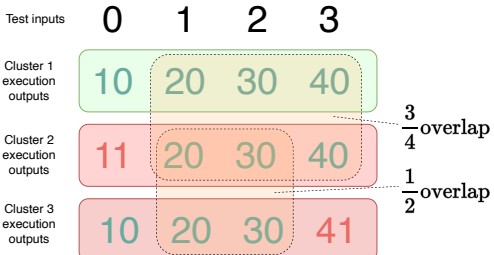

Figure 1: Illustration on concept of "functional overlap" among clusters of solutions. Cluster 1 outputs [10,20,30,40]. Cluster 2's output is [11,20,30,40]. Cluster 3's output is [10,20,30,40]. As a result, Cluster 1 overlaps Cluster 2 on three values [20,30,40], indicating that they are 3/4 overlapped. Cluster 1 overlaps Cluster 3 on three values [10,20,30], which can also be considered 3/4 overlapped. Cluster 1 has a functional overlapping score of $3 + 3 = 6$. Cluster 2 overlaps with Cluster 3 on two values [20,30], resulting in a functional overlapping score of $2 + 3 = 5$, and Cluster 3 has a functional overlapping score of 5. Thus, Cluster 1 has the highest cumulative functional overlap, is most representative and likely to be the optimal solution.

For instance, AlphaCode (Li et al., 2022) uses a distinct model to create test inputs. Solutions are then executed against these inputs, and those with matching outputs are clustered. This is based on the understanding that while there can be numerous incorrect program variations, correct ones often show similar patterns, leading to clusters being ranked by their solution count.

Conversely, CodeT (Chen et al., 2023) clusters solutions that clear the same model-generated test cases. However, this can group functionally diverse solutions. If a cluster's solutions only pass some test cases, it's uncertain if the outputs for the failed cases are consistent across solutions, potentially compromising cluster functionality and the confidence in selecting from these ranked clusters.

## 2.3 MODELING INTER-CLUSTER RELATIONSHIPS

Existing clustering and reranking methods analyze clusters independently without inter-cluster relationships (Li et al., 2022; Chen et al., 2023). However, modeling these interactions can better indicate cluster correctness. As such, we introduce a new metric called "functional overlap" to quantify cluster similarity based on execution outputs, as shown in Figure 1. We can execute code solutions from each cluster on the same test inputs and compare their outputs. The level of output match indicates the extent of functional overlap between two clusters.

The intuition is that clusters with high overlap exhibit greater shared functionality. By modeling the extent to which a cluster overlaps with others, functional overlap identifies the most "representative" cluster. A cluster with maximal cumulative overlap has outputs most consistent with all other clusters. As inconsistencies often indicate incorrect functionality, the cluster interacting most comprehensively is likely the optimal solution. This is similar to the assumptions of Fischler & Bolles (1981) where incorrect solutions are diverse and there is a low probability of having a functional agreement among incorrect solutions.

## 3 APPROACH DETAILS

### 3.1 OVERVIEW

Figure 2 provides an overview of our end-to-end approach. First, given a well-trained CodeLLM (such as Codex) and three inputs: (1) Task description, (2) Code generation prompt, (3) Test case generation prompt, we instruct the CodeLLM to generate a set of code solutions as well as test cases. Specifically, we prompt the CodeLLM to produce a collection of solutions $\mathbf{S} = \{s_1, s_2, ..., s_N\}$ and a set of test cases $\mathbf{T} = \{t_1, t_2, ..., t_M\}$, where $N$ and $M$ are hyperparameters defining the number of solutions and test cases.

Each test case $t_i$ consists of two components: the test input $z_i$ and the expected output $\hat{o}i$ based on the context (e.g. `assert add(1,2) == 3`, where (1,2) is the input and 3 is the output). We can

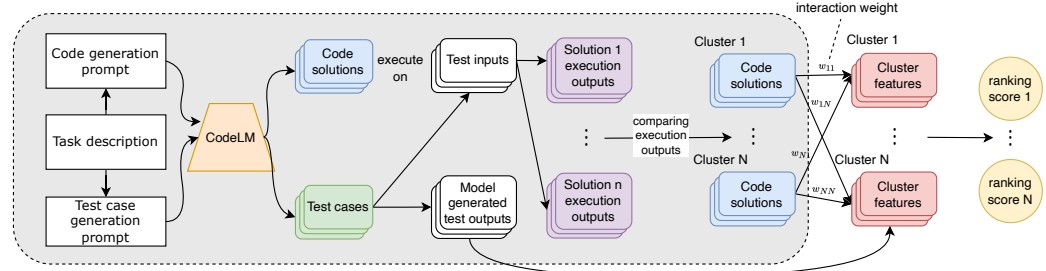

Figure 2: Method overview.

then execute the test inputs $\mathbf{Z} = \{z_1, z_2, ..., z_M\}$ on the set of solutions $\mathbf{S}$ to generate the execution outputs $\mathbf{O} = \{o_{11}, o_{12}, ..., o_{NM}\}$.

Next, we cluster the solutions $\mathbf{S}$ into groups $\mathbf{C} = \{C_1, C_2, ..., C_K\}$ based on their execution outputs, where $K$ is the number of unique clusters. We then compute an interaction matrix $\mathbf{I}$ to quantify the functional overlap between clusters.

Finally, we multiply the interaction matrix $\mathbf{I}$ by a validation score vector $\mathbf{V}$ to obtain final ranking scores $\mathbf{R}$ for selecting the optimal solutions. The validation scores in $\mathbf{V}$ represent features of each cluster, such as the number of solutions.

In the following sections, we elaborate on the key steps of our algorithm.

## 3.2 CLUSTERING SOLUTIONS BY EXECUTION OUTPUTS

We first execute each solution $s_i \in \mathbf{S}$ on the test inputs $\mathbf{Z}$ to produce execution outputs $\mathbf{O}$. Solutions that exhibit identical execution outputs are grouped into the same cluster: $F(s_i) = F(s_j) \iff \mathbf{O}s_i = \mathbf{O}s_j$.

Here, $F$ represents the clustering function that maps a solution $s$ to a cluster identifier $k$. The above equation indicates that two solutions $s_i$ and $s_j$ are assigned to the same cluster if and only if their output sets $\mathbf{O}s_i$ and $\mathbf{O}s_j$ are exactly equal.

## 3.3 COMPUTING INTERACTION MATRIX

After obtaining execution outputs $o_{ij}$ for each cluster $C_i$ on test input $z_j$, we define an interaction matrix $\mathbf{I} \in \mathbb{R}^{K \times K}$ to quantify functional overlap:

$$I_{ij} = \frac{1}{M} \sum_{k=1}^{M} \delta(o_{ik} = o_{jk}) \tag{1}$$

Here, $o_{ik}$ and $o_{jk}$ refer directly to the execution outputs of clusters $C_i$ and $C_j$ respectively on the $k^{\text{th}}$ test input. $\delta$ is the indicator function that returns 1 if the condition inside is true and 0 otherwise.

## 3.4 COMPUTING FINAL RANKING SCORES

In addition to modeling inter-cluster interactions via $\mathbf{I}$, we also consider an extra validation dimension $\mathbf{V} \in \mathbb{R}^{K \times 1}$ containing cluster features. For instance, $V_i$ could represent the number of solutions in cluster $C_i$ (abbreviated as cluster sizes) or the number of test cases that the solutions in cluster $C_i$ passed (abbreviated as pass rates), providing a notion of cluster confidence. The final ranking vector $\mathbf{R} \in \mathbb{R}^{K \times 1}$ can then be computed as $\mathbf{R} = \mathbf{I} \cdot \mathbf{V}$. Here, $R_i$ aggregates information about both the inter-cluster interactions of $C_i$ (via $\mathbf{I}$) and its cluster features (via $\mathbf{V}$). Clusters with higher ranking scores in $\mathbf{R}$ are those with significant functional overlap to other clusters and high validity according to $\mathbf{V}$. By considering inter-cluster relationships and functional similarity in a principled manner, we believe our ranking approach can effectively identify the most promising solutions. We validate our method through extensive experiments in the following sections.

## 4 EXPERIMENTAL SETUP

**Models** We evaluate our method on several state-of-the-art CodeLLMs, including Codex, Wizard-Coder, StarCoder and CodeGen. Each model family has different model size (e.g. WizardCoder 15B and 34B) and different training methods (e.g. base model and instruction fine-tuned model). As a result, we chose a diverse set of models, ranging from small to large scale, and from base model to instruction fine-tuned model, to demonstrate the efficacy of our ranking method on a diverse set of models. In total, we demonstrate our approach on 9 models ranging from 6B to 34B parameters.

**Metrics** We use pass@k (Chen et al., 2021), which is often employed to evaluate the functional correctness of code solutions based on code execution instead of similarity-based metrics. A task is considered solved if there exists at least a solution in $k$ sampled solutions that passes all the pre-defined set of test cases. pass@k is the fraction of solved tasks when sampling $k$ solutions from a pool of $n$ solutions where $c$ solutions are correct (with $n \geq c, k$) for each task. Formally, pass@k can be defined as:

$$\text{pass@}k := \mathop{\mathbb{E}}_{\text{Problems}} \left[ 1 - \frac{\binom{n-c}{k}}{\binom{n}{k}} \right] \tag{2}$$

**Baselines** We compare SRank with recent methods for solution reranking, including Coder-Reviewer (Zhang et al., 2023) and CodeT (Chen et al., 2023). Coder-Reviewer (Zhang et al., 2023) is the state-of-the-art method. In the other hand, CodeT (Chen et al., 2023) shares similar clustering-reranking approach to our method. We refer directly to the number reported in CodeT and Coder-Reviewer to compare with SRank. However, since our goal is to provide a comprehensive evaluation on different reranking methods,

**Benchmarks** We use two popular benchmarks in code generation: HumanEval (Chen et al., 2021) and MBPP (sanitized version) (Austin et al., 2021). HumanEval is a dataset for human evaluation of code generation that consists of 164 original programming problems that assess language comprehension, algorithms, and simple mathematics. MBPP is a code generation benchmark dataset made up of approximately 1000 crowd-sourced Python programming problems designed to be solved by entry-level programmers, covering programming fundamentals, standard library functionality, and so on. To avoid exposing real test cases to the language model, we follow the prompt design in Chen et al. (2021) by removing all example input-output cases from context c before generating code solutions and test cases.

**Implementation Details** For Codex002 and CodeGen16B, we refer to the artifact, including both solutions and test cases, provided by Chen et al. (2023). Regarding the remaining models, we use the HuggingFace library a(Wolf et al., 2019) and load models in half-precision. We set the temperature to 0.8, the top $p$ to 0.95, the maximum new tokens to 2048, and the timeout of executing solutions to 5 seconds. For each problem, we sample 100 solutions and 100 sequences of test cases, each sequence can contain multiple test cases. For post-processing solutions and test cases, we follow Chen et al. (2023) to truncate the generated sequences by the five-stop words: "\nclass", "\ndef", "\n#", "\nif", and "\nprint".

## 5 EXPERIMENTAL RESULTS

Table 1 presents the pass@1 results on the HumanEval and MBPP benchmarks on various CodeLLMs. Our method, SRank, consistently outperforms other techniques across most models. For instance, on the HumanEval benchmark, compared to CodeT and Coder-Reviewer, SRank achieves an average improvements over CodeT and Coder-Reviewer of about 3.63%, and 8.81% of pass@1, respectively. Notably, for strong instruction-tuned models, WizardCoder models, CodeT even performs worse than the greedy search approach.

Additionally, when compare Coder-Reviewer with the random sampling method, it is unstable across the models, specifically, using WizardCoder15B and StarCoder as examples, Coder-Reviewer brings modest increases of 4.17% and 6.16% compared to our improvements of 14.79% and 21.44%. On the MPPS benchmark, **SRank** still achieves outstanding performances although the magnitude of

| | HumanEval | | | | | |
|---|---|---|---|---|---|---|
| | WizardCoder34B | WizardCoder15B | CodeGen2.5-Instruct | StarCoder | Codex002 | CodeGen16B |
| Greedy | 68.90 | 50.61 | 28.05 | 39.63 | 47.00 | 29.70 |
| CodeT | 72.36 | 58.64 | 56.81 | 50.51 | 65.80 | 36.70 |
| Coder-Reviewer | - | 49.37 | 45.63 | 38.71 | 66.90 | 42.60 |
| Random | 59.88 | 45.20 | 26.68 | 32.55 | 37.06 | 22.78 |
| SRank | **75.31** | **59.99** | **60.55** | **53.99** | **69.66** | **43.07** |
| | MBPP-S | | | | | |
| | WizardCoder34B | WizardCoder15B | CodeGen2.5-Instruct | StarCoder | Codex002 | CodeGen16B |
| Greedy | 60.42 | 51.29 | 42.86 | 45.90 | 58.10 | 42.40 |
| CodeT | 63.39 | 58.18 | 55.02 | 58.05 | 67.70 | 49.50 |
| Coder-Reviewer | - | 52.52 | 52.74 | 49.48 | 64.70 | 50.30 |
| Random | 54.37 | 45.72 | 34.60 | 39.26 | 47.50 | 31.54 |
| SRank | **64.14** | **59.01** | **57.02** | **58.38** | **69.25** | **51.03** |

Table 1: Results of pass@1 on HumanEval and MBPP-S benchmarks in the zero-shot setting compared to SOTA methods, CodeT and Coder-Reviewer.

| | HumanEval | | | | | |
|---|---|---|---|---|---|---|
| | WizardCoder34B | WizardCoder15B | CodeGen2.5-Instruct | StarCoder | Codex002 | CodeGen16B |
| Cluster sizes | 72.17 | 56.38 | 55.92 | 48.63 | 59.43 | 40.51 |
| Pass rates | 65.09 | 43.07 | 36.17 | 35.28 | 58.37 | 21.89 |
| Cluster sizes + Pass rates | 73.28 | 58.21 | 58.35 | 51.90 | 66.07 | 41.72 |
| Interaction + Cluster sizes | 73.79 | 58.16 | 59.46 | 53.12 | 65.84 | 42.48 |
| Interaction + Pass rates | 73.59 | 53.49 | 48.37 | 51.13 | 65.91 | 34.61 |
| SRank (all) | **75.31** | **59.99** | **60.55** | **53.99** | **69.66** | **43.07** |
| | MBPP-S | | | | | |
| | WizardCoder34B | WizardCoder15B | CodeGen2.5-Instruct | StarCoder | Codex002 | CodeGen16B |
| Cluster sizes | 65.46 | 56.17 | 56.76 | 55.50 | 64.22 | 53.00 |
| Pass rates | 61.88 | 49.93 | 43.80 | 48.57 | 60.80 | 36.67 |
| Cluster sizes + Pass rates | 64.38 | 58.13 | **57.30** | 57.68 | 68.78 | 50.60 |
| Interaction + Cluster sizes | **66.39** | 56.80 | 55.61 | 55.58 | 66.50 | **53.08** |
| Interaction + Pass rates | 63.33 | 56.11 | 50.27 | 51.33 | 64.53 | 42.78 |
| SRank (all) | 64.14 | **59.01** | 57.02 | **58.38** | **69.25** | 51.03 |

Table 2: Results of Ablation Study on combining different cluster features

improvements is slightly less than that of HumanEval. Our comprehensive experiments demonstrate the effectiveness of **SRank** over CodeT and Coder-Reviewer.

## 5.1 ALATION STUDY

In this section, we perform a few ablation studies to measure the impact of different features on the performance of our method.

**Impact of Cluster Features**   We aim to evaluate performance when the ranking scores are solely based on cluster features, which can be cluster sizes, pass rates, or a combination of the two. The Table 2 shows SRank's performance when these features are added or removed from the ranking pipeline. When ranking the solutions using only one of them, we can see that *Cluster sizes* are more important than *Pass rates*. However, this does not mean that *Pass rates* are unimportant. When both features are combined, the results can be improved even further. Finally, we achieve the best SRank performance by combining both of these features or one of them with the Interaction matrix.

**Impact of Interaction Matrix**   We conduct additional experiments to demonstrate the effectiveness of the interaction matrix $\mathbf{I}$. From the results in Table 2, it is obvious that the interaction matrix $\mathbf{I}$ helps to boost the performance for any cluster features. Importantly, integrating *cluster sizes* with the $\mathbf{I}$ achieves results on par with CodeT, highlighting the significance of interactions among clusters.

|  | HumanEval | MBPP-S |
|---|---|---|
| N.Coder | 57.15 | 56.73 |
| Interaction + N.Coder | 62.69 | 59.50 |
| N.Reviewer | 55.33 | 55.41 |
| Interaction + N.Reviewer | 62.69 | 59.74 |
| N.CoderReviewer | 62.67 | 59.39 |
| Interaction + N.CoderReviewer | **63.30** | **61.38** |

Table 3: CoderReviewer ranking criterions as cluster fearures.

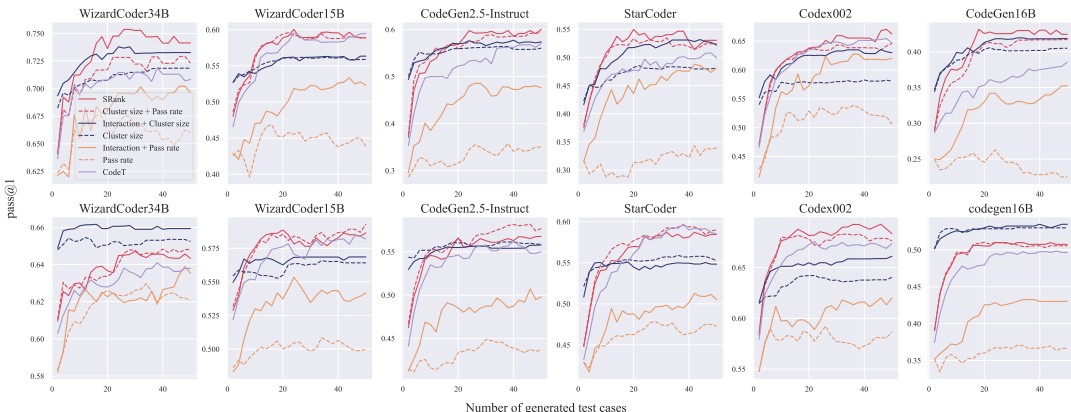

Figure 3: Ablation study on scaling number of model generated test cases vs. pass@1. The first row of figures presents the performance in HumanEval, while the second row presents results of MBPP

In addition, Figure 4 depicts changes in cluster rank. The orange solid lines indicate cluster ranks when **I** (W-I) is involved, while the blue solid lines indicate cluster ranks when **I** (W/O-I) is not involved. We can see that the ranks of valid clusters (clusters that contain valid solutions) are usually higher than W/0-I. This demonstrates the significance of **I** in our ranking strategy.

**Scaling Number of Generated Test Cases**  Our method necessitates generating test inputs for clustering and full test cases to determine the pass rate, which increases time and resource consumption. We conducted an ablation study to assess how the number of test cases influences code generation performance. Figure 3 shows the pass@1 performance based on the number of test cases generated, ranging from 2 to 50.

Comparing solid (with interaction matrix) and dashed lines (w/o interaction matrix), cluster interaction consistently enhances performance over solely using cluster features. The performance gap between methods with and without interaction varies by CodeLLM and test case count, with larger improvements as test cases increase, showcasing our method's scalability. However, with limited test cases, SRank sometimes underperforms due to each ranking feature's potential negative impact when interaction is considered. For optimal results with SRank, we suggest generating at least 30 test cases to fully benefit from cluster interaction, feasible within 1 to 2 sampling rounds given current language models' context lengths.

Additionally, comparing the performance of CodeT with our method reveals disparities in some CodeLLMs. This discrepancy arises from clustering quality. Limited, low-quality test cases can lead to semantically inconsistent clusters, making ranking them less meaningful. In contrast, clustering by execution outputs ensures functional consistency, enhancing ranking reliability.

**CoderReviewer Ranking Criteria as Cluster Features**  We investigate adding cluster interaction with CoderReviewer ranking criteria as cluster features, where Coder represents the likelihood $P_{CodeLLM}(x|y)$ and Reviewer represents $P_{CodeLLM}(y|x)$. CoderReviewer is the multiplication $P_{CodeLLM}(x|y)P_{CodeLLM}(x|y)$, with $x$ as the task description and $y$ as generated code solution.

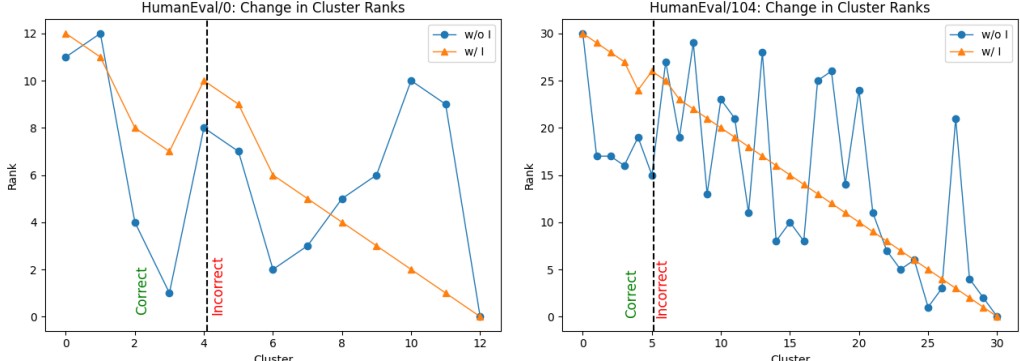

Figure 4: Demonstration of the impact of the interaction matrix on cluster ranking. The two graphs represent two problems chosen from Human-Eval, with solutions generated by CodeGen2.5-Instruct.

| | # Problem: HumanEval/45
def triangle_area(a, h):
"""Given length of a side and high
return area for a triangle""" | # Problem: HumanEval/106
def f(n):
"""Implement the function f that takes n as a parameter,   and returns a list
of size n, such that the value of the element at index i is the factorial of i if i
is even   or the sum of numbers from 1 to i otherwise. i starts from 1.""" | |
|---|---|---|---|
| **CodeT** | `area = 0.5 * a * h`
`return area` #1 | `result = []`
`for i in range(1, n + 1):`
`  if i % 2 == 0:`
`    result.append(i)`
`  else:`
`    result.append(sum(range(1, i +`
`1)))`
`return result` #1 | `result = [None] * n`
`for i in range(1, n+1):`
`  if i % 2 == 0:`
`    result[i-1] = i * (i-1)`
`  else:`
`    result[i-1] = sum(range(1,`
`i+1))`
`return result` #2 |
| | `area = (1/2) * a * h`
`return round(area, 2)` #2 | | |
| | `return 0.5 * a * h` #3 | | |
| **SRank** | `area = 0.5 * a * h`
`return area` #4 | `result = []`
`for i in range(1, n + 1):`
`  if i % 2 == 0:`
`    result.append(i)`
`  else:`
`    result.append(sum(range(1, i +`
`1)))`
`return result` #3 | `result = []`
`for i in range(1, n+1):`
`  if i % 2 == 0:`
`    result.append(i)`
`  else:`
`    temp = 0`
`    for j in range(1, i+1):`
`      temp += j`
`    result.append(temp)`
`return result` #4 |
| | `area = (a * h) / 2`
`return area` #5 | | |
| | `return 0.5 * a * h` #6 | | |

Figure 5: Case studies from HumanEval's problems with the highest-score clusters produced by CodeT versus SRank using CodeGen2.5-Instruct.

The prefixed N. before each term indicates that the likelihood is normalized by the number of sequence tokens, which was shown to improve performance in previous work. The cluster feature in this cases is the criteria score of the solution with maximum score within that cluster. Experimental results in Table 3 show consistent improvement when adding cluster interaction over Coder-Reviewer. This demonstrates that when the interaction matrix is combined with CoderReviewer, the interaction matrix can improve CoderReviewer's performance even further, indicating that our method is adaptable to different reranking methods.

## 5.2 CASE STUDY

In this section, we compare the quality of cluster samples from CodeT and our method. The top 1 clusters ranked by CodeT and SRank are shown in Figure 5. It is clear that functional inconsistency among solutions plagues CodeT's clusters. For example, in the problem HumanEval 45, solution #2 is not semantically equivalent to solutions #1 and #3, and in the problem HumanEval 106, solution #1 differs from solution #2. This phenomenon is explained by the fact that CodeT groups solutions that pass the same set of test cases into clusters. As a result, when computing the pass@k, inconsistency in functionalities among top cluster solutions can degrade performance. *SRank*, on the other hand, considers execution outputs for the clustering process, ensuring functional consistency among solutions within a cluster.

## 6 RELATED WORK

**Code Large Language Models** The advent of large language models has revolutionized code understanding and generation tasks (Rozière et al., 2023; Li et al., 2023a; Nijkamp et al., 2023a; Wang et al., 2023; Nijkamp et al., 2023b; Fried et al., 2023; Li et al., 2023b), including code comprehension (Bui et al., 2021c;b;a; Mou et al., 2016), code generation (Feng et al., 2020; Wang et al., 2021; Elnaggar et al., 2021), code completion (Feng et al., 2020; Wang et al., 2021; Peng et al., 2021), program repair (Xia et al., 2022; Bui et al., 2022). Recent research has leveraged natural language processing models for code tasks, adopting pretraining strategies similar to those for natural languages (Feng et al., 2020; Wang et al., 2021; Guo et al., 2020; Ahmad et al., 2021; Bui et al., 2021b; Elnaggar et al., 2021; Peng et al., 2021; Kanade et al., 2020; Chakraborty et al., 2022; Ahmed & Devanbu, 2022; Niu et al., 2022). For example, CodeBERT (Feng et al., 2020) adapts a Roberta model (Liu et al., 2019) for multiple programming languages, while CodeT5 (Wang et al., 2021) uses unique identifier information to pretrain the T5 model (Raffel et al., 2019). Despite their capabilities, the effectiveness of these models in capturing intricate code semantics remains a topic of discussion. In contrast, models integrating code-specific features as inductive biases exhibit a deeper understanding (Allamanis et al., 2018; Hellendoorn et al., 2020).

Although there are numerous CodeLLMs, not all of them are suitable and perform well in terms of code generation; however, large enough models are the best models. StarCoder (Li et al., 2023a), trained on a vast dataset and handling contexts up to 8,000 tokens; CodeLlama (Rozière et al., 2023), with a capacity for 100,000 tokens; and WizardCoder (Luo et al., 2023), fine-tuned for evolving code instructions. We also delve into models like CodeGen2.5-Instruct (Nijkamp et al., 2023a), CodeT5+Instruct (Wang et al., 2023), Codex002 (Chen et al., 2021), CodeGen1-Mono 16B (Nijkamp et al., 2023b), and InCoder 6B (Fried et al., 2023).

**Reranking Methods for Code Generation** Numerous studies have focused on the reranking code generated by language models Chen et al. (2021); Zhang et al. (2023); Ni et al. (2023); Chen et al. (2023); Inala et al. (2022). These works primarily prioritize solutions drawn from language models. Chen et al. (2021) empirically demonstrated that selecting solutions based on the mean log probability of tokens improves performance. Coder-Reviewer (Zhang et al., 2023) proposed a mutual information-based ranking method for natural language instructions and generated solutions. Reranking has also been done using execution-based metrics. MBR-exec (Shi et al., 2022) seeks to minimize a loss function across all solutions, whereas AlphaCode citepli2022competition clusters solutions based on execution outputs.LEVER citepni2023lever uses a verifier to assess the correctness of a program, whereas CodeT (Chen et al., 2023) leverages the capability of generating high-quality test cases. Our approach is distinct in that it does not require model training or fine-tuning and can complement methods such as LEVER Ni et al. (2023). Future research could benefit from integrating both methods.

## 7 CONCLUSION

We proposed SRank, a novel reranking strategy for obtaining the best code generation solutions from CodeLLMs. **SRank** focuses on *modeling inter-cluster relationships*, with the goal of identifying the cluster with the most "*functional overlap*" with other clusters. This allows for the identification of the most representative cluster with a significant overlap with all other clusters. Because inconsistencies frequently indicate incorrect functionality, the cluster interacting most comprehensively is most likely the best solution. We can better identify the most promising solution by incorporating these inter-cluster relationships into the ranking pipeline.

We demonstrated that **SRank** achieved state-of-the-art performance on pass@1 on a wide range of well-known CodeLLMs when compared to other ranking methods, such as CodeT and Coder-Reviewer, through extensive evaluations. Our ablation study also demonstrates that our method is more applicable to realistic scenarios with a limited number of test cases. Even if the number of test cases is limited, we can still find the best solutions. We believe that this finding is significant because the code generation scenario in real-world use cases is much more difficult than the evaluation benchmarks we are using, and it sheds light on how to choose better solutions given limited constraints (e.g. test case) in the coding environment.

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

# A  VALIDITY OF OUR ASSUMPTION

## A.1  QUANTITATIVE ASSESSMENT

As delineated in Section 2.3, our proposed method, SRank, operates under an assumption that *incorrect solutions are diverse and there is a low probability of functional agreement among incorrect solutions*. Beyond conducting a comprehensive evaluation to establish the superiority of our method over others, it is imperative to validate this foundational assumption. Formally, we introduce some notations for this purpose. Let $\mathbf{F}$ denote the set of solutions sampled from a certain CodeLLM, $s_i$ be the i-th solution in $\mathbf{F}$, $\mathbf{C_k}$ be the k-th cluster by our clustering algorithm, $C^i$ be the cluster including the solution $s_i$, and $|\mathbf{C}_k|$ and $|\mathbf{C}_k|^*$ be the number of solutions and quantity of incorrect solutions in the k-th cluster, respectively. Additionally, $s_i = s_j$ is defined as the two solutions sharing the same semantic functionality, while $s_i \neq s_j$ denotes the opposite. The function $f(s_i, s_j)$ is defined as the computation of the functional overlap between $s_i$ and $s_j$, similar to the Eq. 1. We then compute the probability of having two equivalently semantic solutions below:

$$p(s_i = s_j) = \frac{\sum_k \binom{|C_k|}{2}}{\binom{|F|}{2}} \tag{3}$$

Given that our clustering algorithm ensures all the solutions within a cluster share functionality, and solutions from distinct clusters are certainly not equivalently semantic, we just need to randomly select two solutions from a cluster to satisfy $s_i = s_j$. Thus, the probability of having two equivalently

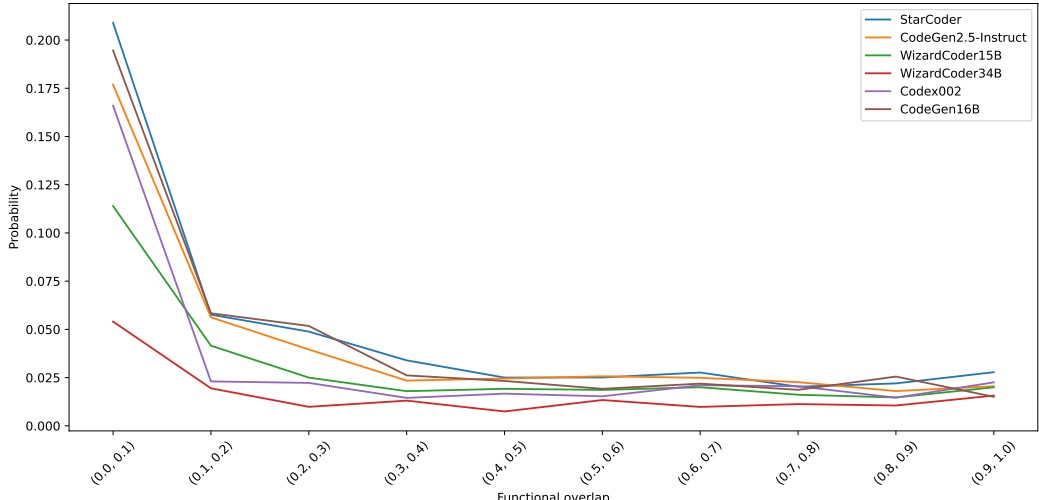

Figure 6: Probability of incorrect solutions varied based on the degree of functional agreement on HumanEval.

semantic incorrect solutions is

$$p(s_i = s_j, s_i \text{ and } s_j \text{ are incorrect}) = \frac{\sum_k \binom{|C_k|^*}{2}}{\binom{|F|}{2}} \qquad (4)$$

Subsequently, we calculate the probability of incorrect solutions with varying levels of functional overlap.

$$p(\text{l} \le f(s_i, s_j) < \text{h}, s_i \ne s_j, s_i \text{ and } s_j \text{ are incorrect}) = \frac{\sum_{(i,j) \in \mathcal{M}} |C^i|^* |C^j|^*}{\binom{|F|}{2}} \qquad (5)$$

Here, $\mathcal{M}$ is comprised of pairs of $(s_i, s_j)$ where $\text{l} \le f(s_i, s_j) < \text{h}$, and l and h are the two hyperparameters.

We consider two range values, $(l_1, h_1)$ and $(l_2, h_2)$ with the same length, where $l_1 < l_2$ and $h_1 < h_2$. According to our assumption, we anticipate that the following inequality holds:

$$p(\text{l}_1 \le f(s_i, s_j) < \text{h}_1, s_i \ne s_j, s_i \text{ and } s_j \text{ are incorrect}) > p(\text{l}_2 \le f(s_i, s_j) < \text{h}_2, s_i \ne s_j, s_i \text{ and } s_j \text{ are incorrect}) \qquad (6)$$

The left term denotes the probability of lower functional agreement among incorrect solutions, while the right term signifies the corresponding probability of higher functional agreement among incorrect solutions. The observed relationship indicates that the probability of lower functional agreement is greater than that of higher functional agreement, substantiating our assumption. Results in Figure 6 illustrate a decline in probability with increasing values of $l$ and $h$. Particularly for the range $(l, h) = (0, 0.1)$, the probability is significantly higher compared to those of other ranges, and at the next range, $(l, h) = (0.1, 0.2)$, the probability experiences a notable decrease. These trends align with our assumption. Moreover, results in Table 4 demonstrate that excluding WizardCoder34B, the probabilities of having two equivalently semantic solutions are much less than 0.5, implying that it is easier to identify two distinct solutions than two solutions sharing the functionality. Additionally, the probability of having two equivalently semantic incorrect solutions is small; specifically, it is three times lower than that of having two equivalently semantic solutions. Consequently, there is a low probability of having two equivalently semantic incorrect solutions.

## A.2 QUALITATIVE ASSESSMENT

Besides quantitative evaluation supporting our assumption, we offer qualitative examples through interaction matrices of solutions generated by StarCoder on HumanEval, as shown in Figure 7. Each

| Model | $p(s_i = s_j)$ | $p(s_i = s_j, s_i \text{ and } s_j \text{ are incorrect})$ |
|---|---|---|
| StarCoder | 0.2176 | 0.0559 |
| CodeGen2.5-Instruct | 0.2692 | 0.0737 |
| WizardCoder15B | 0.4596 | 0.1134 |
| WizardCoder34B | 0.5689 | 0.0990 |
| Codex002 | 0.3218 | 0.0807 |
| CodeGen16B | 0.3482 | 0.1279 |

Table 4: Probabilities of two equivalently-semantic solutions and two equivalently-semantic incorrect solutions among different models

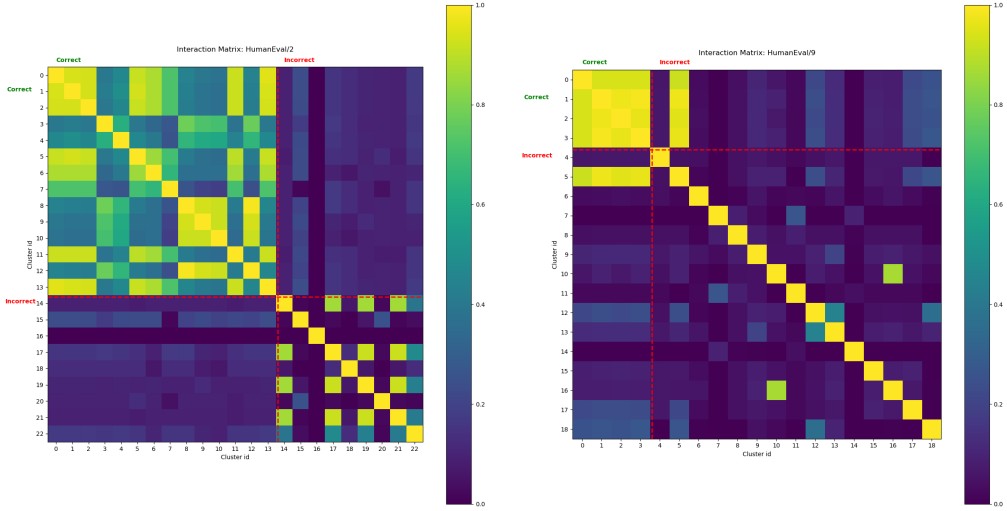

Figure 7: Demonstration of our assumption shows there is a low functional agreement among incorrect solutions. The two graphs represent two problems chosen from Human-Eval, with solutions generated by StarCoder.

matrix is divided into four separate patches by two red-dashed lines: top-left, top-right, bottom-left, and bottom-right. The left region of the vertical line represents clusters, including correct solutions, while the right region encompasses incorrect clusters; similarly, the top region above the horizontal line comprises correct solutions. The diagonal lines are disregarded as their values represent the self-interaction of clusters. It is clearly seen that the top-left patches are notably brighter, whereas the bottom-right patches are virtually dark. These observations demonstrate that correct clusters interact with each other comprehensively, indicating a high probability of functional overlap between correct solutions, while the opposite holds true for incorrect solutions.

