# OpenReview forum: "Neural Rankers for Code Generation via Inter-Cluster Modeling"
_ICLR.cc/2024/Conference — Submitted to ICLR 2024_

### Official Review · Reviewer_53rB · 2023-10-27

**Soundness:** 3 good
**Presentation:** 4 excellent
**Contribution:** 2 fair
**Rating:** 5
**Confidence:** 3

**Summary:**

The paper studies ranking of generated code that are sampled from LLMs. The main idea is to exploit inter-cluster modeling that is overlooked in the literature. The idea is simple - use the overlap of answer with other clusters as the boost signal, assuming that highest overlap is most representative and likely to be the optimal solution. Experiments are done on HumanEval and MBPP-S datasets with 5 code models. Overall the proposed method can outperform 2 baselines published in the literature.

**Strengths:**

S1: the proposed method is simple. It is basically a statistical post-processing method that does not involve LLMs/models.


S2: Experimenting on 6 code models looks comprehensive.

**Weaknesses:**

W1: It is easy to come up with counterexamples to invalidate the method. For example, cluster 1 shares most solutions, but all solutions are wrong. Thus one can at least question the robustness of the proposed solution. The reviewer understands that this probably did not happen often on the datasets tested. But the reviewer can foresee possibilities: for example, when the tasks are difficult and the model may make wrong outputs. Also, the wrong answers match with each other may not be scarce, since the programs are sampled from the same LLMs.


W2: the method is specific to code cluster ranking. The significance of the results are not very clear without a significance test. The reviewer is not sure if it is a good idea to discuss improvement of average performance across various models. With comparing with CodeT, on the 34b model, the improvement seems incremental, and it seems confusing that CodeT is even worse than Greedy.

**Questions:**

Related to W1: any *factual* misunderstanding?

How to understand the significance of the results?

Please proofread the paper. Some sentences do not look complete.

---

> ### Author Response · Authors · 2023-11-21
>
> Dear Reviewer 53rB,
>
> We appreciate the time and effort you have invested in reviewing our manuscript. We try to address your concerns below.
>
> ***Q1. “It is easy to come up with counterexamples to invalidate the method”***
>
> ***A1***:
>
> To ensure the efficacy of our method, we operated under the assumption that incorrect solutions exhibited diversity with a low probability of functional agreement among them. This functional assumption also aligned with the practices of AlphaCode [1] and CodeT [2]. However, in certain cases, this assumption may be untenable as LLMs virtually generate incorrect solutions and erroneous test cases. The scarcity of correct solutions may be scarce compared to incorrect ones introduces  vulnerability to criteria based on sizes of clusters [1,2], especially when incorrect solutions may surpass correct ones in magnitude. In addition, the prevalence of erroneous test cases undermines the reliability of execution-based criteria [2]. These occurences hinder the effective re-ranking of clusters, causing both SRank and CodeT to falter in their re-ranking capabilities.
>
>
> ***Q2. “The significance of the results are not very clear without a significance test.”***
>
> ***A2***:
>
> Given a predetermined set of test cases and corresponding solutions, CodeT and our proposed method, SRank, are deterministic, signifying an absence of randomness that could impact their return values. In order to establish our method’s superiority over CodeT, a comprehensive evaluation was undertaken.
>
> Initially, we assessed 6 models across 2 standard benchmarks, including HumanEval and MBPP, which indicated that SRank always obtained better results than CodeT. The extent of performance disparity between SRank’s performance and that of CodeT varied depending on the model and the dataset. For instance, SRank could achieve 6.37\% more than CodeT in pass@1 on CodeGen16 and HumanEval, marking a substantial 17.36\% improvement while the improvement on WizardCoder34B and HumanEval was 2.95\% in pass@1 (~4.08\%). Therefore, the evaluation spanned multiple models and benchmarks, underscoring the robustness and effectiveness of our proposed method.
>
> Subsequently, we conducted an ablation study to analyze the impact of the number of test cases on the performance. Results in Figure 3 showed that when the number of test cases varied, SRank consistently performed better than CodeT.
>
> ***Q3. “CodeT is even worse than Greedy”***
>
> ***A3***:
>
> Please allow us to explain that CodeT is worse than Greedy, an observation rooted in the distinct prompts employed for each. In the case of Greedy, our prompting instruction fed to LLMs includes test cases sourced from the datasets, functioning as soft constraints that guide LLMs behavior. Furthermore, the instruction-tuned models, like the WizardCoder family, exhibit a profound comprehension of natural language prompts. Consequently, they generate solutions cognizant of the test cases in the prompts, thereby enhancing the likelihood of satisfying the test cases. Conversely, when generating solutions for CodeT and SRank, the prompts do not contain test cases, rendering their outputs more susceptible to errors. To sustantiate this assertion, we present experimental results where prompts do not include test cases.
>
> |           | |WizardCoder34B | WizardCoder15B |
> |-----------|-----------|----------------|----------------|
> | HumanEval |Greedy| 68.90  | 50.61   |
> | HumanEval | CodeT|72.36| 58.64|
> | MBPP      | Greedy|60.42 | 51.29  |
> |MBPP|CodeT|63.39|58.18|
>
> The above results illustrate that when employing identical prompts devoid of explicit test case consideration, CodeT surpasses Greedy in performance.
>
> **References:**
> 1. *Li, Y., Choi, D., Chung, J., Kushman, N., Schrittwieser, J., Leblond, R., ... & Vinyals, O. Competition-level code generation with alphacode. In Science, 378(6624), 1092-1097. 2022.*
> 2. *Chen, B., Zhang, F., Nguyen, A., Zan, D., Lin, Z., Lou, J. G., & Chen, W. Codet: Code generation with generated tests. In ICLR. 2023.*

---

### Official Review · Reviewer_BPa9 · 2023-10-28

**Soundness:** 3 good
**Presentation:** 3 good
**Contribution:** 3 good
**Rating:** 5
**Confidence:** 3

**Summary:**

This work proposes a new rerank strategy which focuses on modeling inter-cluster relationship to select the best solution in code generation. Specifically, they introduce a new metric called functional overlap to quantify the similarity between clusters based on their execution outputs. Experimental results show that this approach achieves remarkable results on pass@1 score across several benchmarks.

**Strengths:**

1. This paper is clearly presented and easy to follow.
2. The method is simple and experimental results show that this method is effective.
3. Ablation study in this paper shows the proposed method can be applied to realistic scenarios with a limited number of test cases.

**Weaknesses:**

1. **Reproducibility**: As this paper did not provide any codes in supplementary materials, we are not sure whether this work can be reproduced.
2. Evaluation metric is limited in this paper since there is only pass@1 score. This paper can provide more results using metrics like pass@5, pass@100, exec@1 and ranked pass@1, as shown in [1].
3. Benchmarks are limited. There are only two benchmarks, i.e., HumanEval and MBPP-S, and these benchmarks contain limited examples. We are not sure whether this method works or not in some large-scale datasets such as APPS[1].

[1] Inala et al., Fault-Aware Neural Code Rankers. NeurIPS 2022.

**Questions:**

1. Please answer the questions in Weaknesses regarding the experiments.
2. Several typos to be fixed:

     a. However, selecting the best solutions from ~~among~~ all possible CodeLLM solutions remains a challenge.

    b. a remarkable results -> remarkable results

3. Could you please provide codes of this work?

---

> ### Author Response · Authors · 2023-11-21
>
> Dear Reviewer BPa9,
>
> Thank you for your thoughtful review and invaluable feedback. Your comments are instrumental in refining our work. We appreciate your positive recognition of the paper presentation, our results, and effectiveness of our method. We fixed the typos in the revised manuscript according to your suggestion. We address your questions below.
>
> ***Q1. Reproducibility***
>
> ***A1***:
>
> Please refer to our response in the authors’ Overall Response comment.
>
> ***Q2. “Evaluation metric is limited in this paper since there is only pass@1 score”***
>
> ***A2***:
>
> In the context of real-world usage, it is plausible that end-users typically opt for a singular suggested solution when soliciting the system to generate code. In alignment with this practical scenario, our evaluation protocol centers exclusively around the pass@1 score. This adherence to reporting pass@1 aligns with established conventions observed in numerous works employing HumanEval and MBPP methodologies. Exemplary instances include foundational code language models such as WizardCoder [1], StarCoder [2], and GPT-4 [3], alongside reranking algorithms like Coder-Reviewer [4], LEVER [5], MBR-exec [6], among others.
>
> ***Q3. “Benchmarks are limited.”****
>
> ***A3***:
>
> Please refer to our response in the authors' Overall Response comment.
>
> References:
> 1. *Can Xu, Qingfeng Sun, Kai Zheng, Xiubo Geng, Pu Zhao, Jiazhan Feng, Chongyang Tao, Daxin Jiang. Wizardlm: Empowering large language models to follow complex instructions. 2023.*
> 2. *Raymond Li, Loubna Ben Allal, Yangtian Zi, Niklas Muennighoff, Denis Kocetkov, … StarCoder: May the source be with you! 2023.*
> 3. *OpenAI. GPT-4 Technical Report. 2023.*
> 4. *Zhang, T., Yu, T., Hashimoto, T., Lewis, M., Yih, W. T., Fried, D., & Wang, S Coder reviewer reranking for code generation. In ICML. 2023.*
> 5. *Ni, A., Iyer, S., Radev, D., Stoyanov, V., Yih, W. T., Wang, S., & Lin, X. V. Lever: Learning to verify language-to-code generation with execution. In ICML. 2023.*
> 6. *Shi, F., Fried, D., Ghazvininejad, M., Zettlemoyer, L., & Wang, S. I. Natural language to code translation with execution. In EMNLP. 2022.*

---

### Official Review · Reviewer_6Abb · 2023-10-31

**Soundness:** 3 good
**Presentation:** 3 good
**Contribution:** 3 good
**Rating:** 6
**Confidence:** 4

**Summary:**

This paper presents a novel re-ranking approach, SRank, for solutions generated by CodeLLM. The authors recognize that previous re-ranking methods often overlooked complex function similarities and interactions among diverse solution clusters, leading to suboptimal outcomes. Consequently, they propose SRank, which focuses more on the interplay among clusters. By quantifying functional overlaps between clusters, SRank provides a more refined code solution ranking strategy. Empirical results demonstrate significant improvements in pass@1 scores on both the Human-Eval and MBPP benchmarks compared to traditional methods.

**Strengths:**

- The approach is simple, innovative, and effectively considers relationships among solution clusters.
- Experimental results are convincing, with ablation study findings further corroborating the method's efficacy.

**Weaknesses:**

- Insufficient descriptions of task scenarios and research value. The authors only briefly reference two previous works when depicting solution clustering and re-ranking tasks, which may confuse the readers, e.g., why not just execute the test cases of the dataset instead of going to generate new test cases? Thus, it is advised to emphasize the generation of new test cases and the need for re-ranking due to the lack of human-designed test cases, as done in CodeT.

- Figure for SRank reveals that result of test case execution will determine the final rank. However, the process for generating suitable test cases is inadequately discussed. For example, are special means employed to ensure the quality of generated test cases? How were the test cases generation prompts designed?

- As mentioned, test case quality can significantly impact re-ranking results. For more complex programming problems, the quality of CodeLLM-generated solutions and test cases may decline. In such cases, SRank's performance changes merit discussion. Therefore, it is suggested the authors validate SRank's effectiveness with more challenging datasets (e.g., Apps and CodeContests).

- Table 2 results indicate Cluster sizes are a more effective feature than Pass rates, which seems counterintuitive as Pass rates more closely resemble the Pass@K metric. An in-depth explanation is recommended. Furthermore, no mention is made of how the authors ordered entries based on Cluster sizes feature—ascending or descending?

**Questions:**

See weaknesses.

---

> ### Author Response · Authors · 2023-11-21
>
> Dear Reviewer 6Abb,
>
> Thank you for your helpful comments and valuable suggestions. Below we answer your concerns.
>
> ***Q1. “...why not just execute the test cases of the dataset instead of going to generate new test cases?...”***
>
> ***A1***:
>
> Please refer to our response in the authors' Overall Response comment.
>
> ***Q2. “...the process for generating suitable test cases is inadequately discussed…”***
>
> ***A2***:
>
> For generating test cases, we followed the CodeT’s pipeline to produce 100 sequences. At each time of sampling, we feed the context (function) and an instruction to LLMs to get a sequence that can contains multiple test cases. We further post-process generated test cases to filter out those with syntactic errors. Below is the template of our designed prompt.
>
> ```
> Below is an instruction that describes a task. Write a response that appropriately completes the request.
>
> ### Instruction:
>     I have this function stub, please generate 50 test cases for this function. The function stub is as follow:
>     ```python
>     {input}
>     pass
>     ```
>     - Each test case is in the form of assertion statement, for example: assert {entry_point}(...) == ...
>     - Each test case is in a single line
>     - The length of each test case should be too long, ideally less than or equal to 150 letters
>     - The test input should not be too long
>     - The inputs of test cases should be diverse and cover corner cases of the function
>     - Test cases should not be repeated
>
> ### Response:
>    Here are 50 test cases for function `{entry_point}`:\nassert {entry_point}("
> ```
> where `input` is the function header and `entry_point` is the function name.
>
> ***Q3. “...it is suggested the authors validate SRank's effectiveness with more challenging datasets…”***
>
> ***A3***:
>
> Please refer to the overall response for more details.
>
> ***Q4. “Table 2 results indicate Cluster sizes are a more effective feature than Pass rates, which seems counterintuitive as Pass rates more closely resemble the Pass@K metric”***
>
> ***A4***:
>
> We welcome the opportunity to clarify that results in Table 2 demonstrate that Cluster sizes appear to have a bigger impact than Pass rates, which seemingly looks counterintuitive first as Pass rates closely correlates with Pass@K metric. This problem is primarily attributed to the nature of generated test cases. We prompt LLMs to produce test cases and employ some simple heuristics to assess their syntactic validity. However, they may be prone to errors in which their expected outputs may be incorrect, thereby diminishing the Pass Rates. In contrast, the Cluster sizes are not dependent on expected outputs of the test cases. We operate under the assumption that there are many ways for solutions to be incorrect, whereas correct solutions share the same functionality. This assumption results in correct solutions forming larger clusters, with smaller clusters potentially containing incorrect solutions. Therefore, the Cluster sizes are likely to be more robust than Pass rates.
>
> ***Q5. “Cluster sizes feature—ascending or descending”***
>
> ***A5***:
>
> Please allow us to elucidate that we re-rank cluster sizes feature in the descending order since we assume that there are many ways for solutions to be incorrect while correct solutions share the same functionality. Therefore, correct solutions tend to form larger clusters, and smaller clusters may contain incorrect solutions.
>
> **References:**
>
> 1. *Chen, B., Zhang, F., Nguyen, A., Zan, D., Lin, Z., Lou, J. G., & Chen, W. Codet: Code generation with generated tests. In ICLR. 2023*
> 2. *Ni, A., Iyer, S., Radev, D., Stoyanov, V., Yih, W. T., Wang, S., & Lin, X. V. Lever: Learning to verify language-to-code generation with execution. In ICML. 2023*

---

### Official Review · Reviewer_c5Kt · 2023-11-02

**Soundness:** 2 fair
**Presentation:** 3 good
**Contribution:** 2 fair
**Rating:** 3
**Confidence:** 4

**Summary:**

The paper presents a clustering-based solution for code re-ranking. Given a large language model (LLM) generates 100 solutions and 100 sequences of test cases, the proposed approach clusters the solutions by executing them against the test cases. The paper introduces a new metric called functional overlap to quantify the similarity between clusters based on their execution outputs, which allows for identifying the most representative cluster that exhibits maximal overlap with all other clusters. Then the inter-cluster relationships are incorporated into the ranking pipeline, the approach identifies the most promising solutions.

**Strengths:**

- The proposed approach is effective on standard benchmarks.
- Rigorous evaluation and ablation study.

**Weaknesses:**

- Overall, I do not like the main idea of sampling a large number of solutions and then clustering them and identifying the best ones. This does not seem to be a practical solution.
- The proposed solution is straightforward and there is not much scientific or technical contribution (I am not considering it as a weakness though mentioning it here). I am not sure if this paper passes the bar of ICLR. I would prefer to see the work as a short paper in some other venue.
- Many statements written throughout the paper are not well-grounded.
- It is not necessary to comment about CodeT or Code-Reviewer being not comprehensive about experiments to lift this work.

**Questions:**

- I'm afraid I have to disagree with the last paragraph in section 2.2 regarding CodeT. Can you explain - how grouping solutions that pass the same subset of test cases potentially compromises cluster functionality?
- How does modeling inter-cluster interactions better indicate cluster correctness? [section 2.3]
- Why should we rely on the test cases generated by the model? Is there any relation between what model generates a solution to the input problem and the test cases? If yes, does executing the test cases for the generated solution add value? Can we directly compare the test cases sampled from the model?
- Why the evaluation is confined to Python language only? There are many multilingual benchmarks available now.
- "For each problem, we sample 100 solutions and 100 sequences of test cases." - In a real setting (say in an industry setting), is this a good assumption that 100 solutions will be sampled? As a user, I would prefer to get one recommendation in my IDE, for that, why an LLM that has billions of parameters need to generate 100s of solutions?

---

> ### Author Response · Authors · 2023-11-21
>
> Dear Reviewer c5Kt,
>
> We are sincerely grateful for your insightful reviews. Your professional feedback not only guides us in crafting a more comprehensive and competitive paper but also reinforces our enthusiasm, especially regarding your positive acknowledgment of our results on code benchmarks, rigorous evaluation and ablation study. Below we address your questions.
>
> ***Q1. “...  Can you explain - how grouping solutions that pass the same subset of test cases potentially”***
>
> ***A1***:
>
> In addressing your query, we provide a specific illustration. Consider a scenario utilizing the CodeT reranking method, where a cluster encompasses two solutions, denoted as `f_1` and `f_2`, both passing all prescribed test cases with the exception of one particular assertion, `assert f(i) == o`. In such instances, our assurance extends only to the divergence of `f_1(i)` and `f_2(i)` from `o`; yet, it remains plausible that `f_1(i) != f_2(i)`. This observation underscores the distinct functionality of  `f_1` and `f_2`, despite their classification within the same cluster.
>
> The fundamental objective of clustering is to assemble solutions with identical functionality into sets, subsequently ranking these solution sets as opposed to directly assessing an extensive array of individual solutions. CodeT, unfortunately, deviates from this foundational principle. Notably, its inadequacy becomes pronounced, particularly when the quantity of test cases is limited. This leads to the aggregation of functionally disparate solutions into a shared cluster, resulting in suboptimal performance, as elucidated in Figure 3 within our submitted paper.
>
> Our proposed method systematically addresses the issue of functional inconsistency within clusters. If two solutions, `f_1` and `f_2`, coexist in the same cluster, and given the identical test case as described earlier, it is guaranteed that `f_1(i) == f_2(i)`, irrespective of the output value `o`. Essentially, our clustering algorithm yields finer-grained clusters than CodeT concerning functionality. Please refer to section 5.2 Case Study which shows some examples regarding functional inconsistency of CodeT. Consequently, our algorithm enhances the efficacy of the cluster ranking step. This distinction is illustrated in Figure 3, where we compare the *CodeT* line with the *Cluster size + Pass rate* line. Both lines utilize cluster size and pass rate as cluster features, with the disparity lying solely in the method of cluster formation.
>
> ***Q2. “How does modeling inter-cluster interactions better indicate cluster correctness?”***
>
> ***A2***:
>
> As articulated in the paper, "The intuition is that clusters with high overlap exhibit greater shared functionality". We elucidate why modeling inter-cluster interactions enhances the indication of cluster correctness through a concrete example. Consider solving a competitive programming problem where individuals propose solutions, which may be correct or incorrect. Here, clusters 1, 2, and 3 represent distinct solutions by three programmers addressing the same problem, respectively. In practice, incorrect solutions fail because they cover most cases but fail in specific instances. For two people with two incorrect solutions, each may fail for different reasons. In Figure 2, person 2, i.e cluster 2, may overlook the special case with input as 0, while person 3 forgets to cover input as 3. Consequently, when calculating the overlap in functionality, i.e., the match in outputs when executing the solutions on numerous test inputs, the solutions sharing the maximum output overlap are likely optimal.
>
> To further validate that modeling inter-cluster interactions aids in ranking clusters, we present the results of reranking solely by matrix I without any cluster features V. In this scenario, V is a column vector with 1 at every entry. Consequently, the ith entry in R is the sum of functional overlap between the ith cluster and all other clusters.
>
> **HumanEval**
> ||WizardCoder34B|WizardCoder15B|CodeGen2.5-Instruct|StarCoder|Codex002|CodeGen16B|
> |--|----|---|---|--|--|--|
> |Greedy    | **68.90** |**50.61**| 28.49   |  39.63 |   47.00   | 29.70 |
> |Random   | 59.88 |  45.20    |   26.68   |  32.55 |   37.06    | 22.78 |
> |Interaction only| 66.49   |49.79| **51.13** |**50.01**| **62.75** | **31.31**|
>
> **MBPP**
> ||WizardCoder34B|WizardCoder15B|CodeGen2.5-Instruct|StarCoder|Codex002|CodeGen16B|
> |--|----|--|--|--|--|--|
> |Greedy  |**60.42**|51.29|42.86      |45.90|**58.10**|**42.40**|
> |Random  | 54.37 |   45.72 | 34.60    | 39.26 |47.50 | 31.54 |
> |Interaction only| 59.60|**53.94**|**47.70**|**48.85**| 57.49 | 41.98 |
>
> The results demonstrate that incorporating interaction modeling consistently elevated performance beyond random. Furthermore, reranking with interaction alone significantly outperforms the performance of greedy decoding in some cases, such as 51.13 vs 28.49 for CodeGen2.5,  50.01 vs 39.63 for StarCoder, 62.75 vs 47.00 for CodeX002 in HumanEval.

---

> ### Author Response · Authors · 2023-11-21
>
> ***Q3. “Why should we rely on the test cases generated by the model?”***
>
> ***A3***:
>
> For clarification, we remind that in the context of our work, a test case can be decomposed into a test input and a test output. For example, a full test case `assert f(i) == o` has the test input `i` and test output `o`.
>
> Our reliance on the test cases generated by pre-trained language models stems from their capacity to produce syntactically and semantically accurate test cases. These expansive models are trained on extensive code datasets, thereby imbuing them with the capability to generate test cases that are, to a certain extent, syntactically and semantically sound. Syntactic correctness entails adherence to the format `assert function_name(*args, **kwargs) == output` or in specific cases, `assert function_name(*args, **kwargs) is None/True/False`. Semantically, correctness implies that the test input `*args` and `**kwargs` conform to valid argument types, order, and value ranges, while the test output aligns functionally with the input.
>
> We measure the percentage of test cases with valid syntax and semantically correct test inputs.
> This is done by checking the 1) format or structure of the test case by regular expression, 2) test input and 3) test output by compiling them.
>
> **HumanEval**
> ||WizardCoder34B|WizardCoder15B|CodeGen2.5-Instruct|StarCoder|Codex002|CodeGen16B|
> |-|-|-|-|-|-|-|
> |No. Test cases|40,519|79,272|76,802|58,095|64,137|9124|
> |No. Invalid test cases|589|716|2757|3431|3306|1108|
> |Percentage of invalid test cases|1.45|0.90|3.59|5.91|5.15|12.14|
>
> **MBPP**
> ||WizardCoder34B|WizardCoder15B|CodeGen2.5-Instruct|StarCoder|Codex002|CodeGen16B|
> |-|-|-|-|-|-|-|
> |No. Test cases|105,213|206,942|218,538|127,386|132,439|17,598|
> |No. Invalid test cases|1,960|2,170|9,886|10,118|10,624|1,561|
> |Percentage of invalid test cases|1.86|1.02|4.52|7.94|8.02|8.87|
>
> From the table, since the percentage of invalid test cases is very minor, we can confirm the ability of code language models in generating test cases with correct syntax and correct semantics to a certain extent. Note that the reported numbers above are not deduplicated. Although the total number of test cases seem to be large, for each sequence of test cases, we only choose the first 5 valid test cases. With 100 sequences of test cases, the possibly maximum number of test cases for SRank in our experiments is 500 test cases.
>
> ***Q4. “Why the evaluation is confined to Python language only? There are many multilingual benchmarks available now.”***
>
> ***A4***:
>
> We agree that there are multilingual benchmarks available and our method can be generalized to most of the programming languages. In this work, we choose Python as a demonstration language as it was also employed in previous work with the same research scope, e.g., CodeT [1], Coder-Reviewer [2], LEVER [3], MBR-exec [4], CodeRanker [5], and AlphaCode [6]. We leave the experiments for other programming languages for future work.
>
> ***Q5. “In a real setting (say in an industry setting), is this a good assumption that 100 solutions will be sampled? …, why an LLM that has billions of parameters need to generate 100s of solutions?”***
>
> ***A5***:
>
> As we indicated in the introduction, maximum likelihood-based decoding often suffers from a degeneration problem, while the correct solutions of the problem may exist within the sampled solutions from the model distribution. Empirically, numbers in Table 1 (with updated greedy decoding results) showed that greedy decoding results can be much worse than SRank. This supports the argument that sampling a sufficiently enough number of solutions
> can lead to higher chances that those sampled solutions would contain the correct one. Reranking algorithms such as SRank are specifically devised to discern and prioritize the correct solutions among the sampled set. Again, the ultimate goal of this work and many other works e.g CodeT, Coder-Reviewer, LEVER, MBR-exec, CodeRanker, AlphaCode, etc., is to improve performance for real world tasks, in this case, code generation, through reranking sampled solutions with the minimal trade-off for time and resources.
>
> Besides, the number 100 solutions is employed in the paper and can be seen as a hyperparameter. In application, a specific number can be adopted to balance the trade-off between accuracy for time and resources since performance tends to improve as more solutions are sampled.

---

> > ### Author Response · Authors · 2023-11-21
> >
> > References:
> >
> > 1. *Chen, B., Zhang, F., Nguyen, A., Zan, D., Lin, Z., Lou, J. G., & Chen, W. Codet: Code generation with generated tests. In ICLR. 2023.*
> > 2. *Zhang, T., Yu, T., Hashimoto, T., Lewis, M., Yih, W. T., Fried, D., & Wang, S Coder reviewer reranking for code generation. In ICML. 2023.*
> > 3. *Ni, A., Iyer, S., Radev, D., Stoyanov, V., Yih, W. T., Wang, S., & Lin, X. V. Lever: Learning to verify language-to-code generation with execution. In ICML. 2023.*
> > 4. *Shi, F., Fried, D., Ghazvininejad, M., Zettlemoyer, L., & Wang, S. I. Natural language to code translation with execution. In EMNLP. 2022.*
> > 5. *Inala, J. P., Wang, C., Yang, M., Codas, A., Encarnación, M., Lahiri, S., ... & Gao, J. Fault-aware neural code rankers. In NeurIPS. 2022.*
> > 6. *Li, Y., Choi, D., Chung, J., Kushman, N., Schrittwieser, J., Leblond, R., ... & Vinyals, O. Competition-level code generation with alphacode. In Science, 378(6624), 1092-1097. 2022.*

---

> > > ### Comment · Reviewer_c5Kt · 2023-11-22
> > > **Thank you for the detailed response**
> > >
> > > Thank you for addressing my questions. However, I am still not satisfied with responses on the following issues.
> > >
> > > 1. I still didn't understand the point “modeling inter-cluster interactions better indicate cluster correctness”. Basically, my concern is the assumption (section 2.3)- "incorrect solutions are diverse and there is a low probability of having a functional agreement among incorrect solutions" which I thought the authors will clarify. Given we sample solutions using LLMs, it is important to validate why two incorrect solutions would be diverse in terms of functionality?
> > > 2. Why we need to sample 100s of solutions? If we sample only 5-10 solutions, do we still need the proposed approach?
> > > 3. When prior works published, there might not be multilingual benchmarks. Given we have such benchmarks now, current works should start to use them.
> > >
> > > I am retaining my score as I do not convinced that the proposed approach will be useful for real scenario.

---

> > > > ### Author Response · Authors · 2023-11-23
> > > >
> > > > Dear Reviewer c5Kt,
> > > >
> > > > Thank you for your reply.  We are delighted to address your concerns below.
> > > >
> > > > ***Q6. “Given we sample solutions using LLMs, it is important to validate why two incorrect solutions would be diverse in terms of functionality?”***
> > > >
> > > > ***A6***:
> > > >
> > > > To clarify our foundational assumption, we present comprehensive results in the section A of the appendix. Specifically, we conduct a quantitative evaluation by computing the probability of two incorrect solutions with varying levels of functional overlap, and the obtained results robustly support our assumption. Furthermore, we also provide some qualitative examples through interaction matrices that demonstrate that there is high functional overlap among correct solutions, whereas the opposite holds for incorrect solutions. For a more in-depth explanation, please refer to section A in the appendix.
> > > >
> > > >
> > > > ***Q7. “Why we need to sample 100s of solutions? If we sample only 5-10 solutions, do we still need the proposed approach?”***
> > > >
> > > > ***A7***:
> > > >
> > > > In response to the query regarding the necessity of sampling 100 solutions, it is imperative to recognize that an augmented pool of sampled solutions enhances the likelihood of including the correct ones within the set of generated solutions. This augmentation significantly amplifies the efficacy of reranking algorithms, particularly when dealing with a sufficiently large and diverse sample.
> > > >
> > > > Constraining the number of generated solutions to a mere 5-10 introduces a dependency of the proposed approach's performance on specific model characteristics and hyperparameters governing the sampling process, such as temperature, top-k, top-p, and the like. To comprehensively assess the optimal number of solutions required to surpass the performance of greedy decoding, an ablation study is imperative. Unfortunately, due to the time constraints of the rebuttal period, immediate presentation of results is unfeasible. However, a commitment to conducting a thorough experiment on this ablation and incorporating the findings into the subsequent revision is assured.
> > > >
> > > > ***Q8. “When prior works published, there might not be multilingual benchmarks. Given we have such benchmarks now, current works should start to use them.”***
> > > >
> > > > ***A8***:
> > > >
> > > > We express our gratitude to the reviewer for proposing an exploration of alternative benchmarks encompassing diverse programming languages. Acknowledging the evolving landscape, we recognize the existence of contemporary multilingual benchmarks. It is duly noted that the integration of experiments involving additional programming languages is a prospect we are poised to address in future iterations of our work.

---

### Author Response · Authors · 2023-11-21
**Overall Response: Paper changes**

Dear AC and Reviewers,

We want to express our appreciation to the reviewers for their time and expertise in providing detailed and constructive comments that have greatly contributed to the refinement of our paper.

After submission, we revised our manuscript. We have updated the paper as follows:
1. In the introductory paragraph of Section 2.3, the reference to the figure is incorrect. We acknowledge the error in figure indexing and regret any confusion caused. We updated the correct figure index.
2. We have rectified the results of Greedy decoding in Table 1 due to an oversight in prompt construction during the collection of generated solutions with greedy decoding. Throughout our paper, we adhere to the methodology of CodeT [1] for solution generation. In our experiments with HumanEval, the original contexts, containing example input-output cases, are excluded to avoid exposing real test cases. The results presented in the manuscript were unintentionally gathered with contexts still encompassing these test cases. The corrected numbers for greedy decoding under the same prompt conditions are provided below. In summary, without revealed test cases in the prompt, performance consistently diminishes for all models, as anticipated.

**HumanEval**
||WizardCoder34B|WizardCoder15B|CodeGen2.5-Instruct|StarCoder|Codex002|CodeGen16B|
|-----------|-----------|----------------|----------------|--|--|--|
|~~Incorrect Greedy~~|~~73.20~~|~~59.80~~|~~36.20~~|~~40.80~~|~~47.00~~|~~29.70~~|
|*Updated Greedy*|*68.90*|*50.61*|*28.49*|*39.63*|*47.00*|*29.70*|
|CodeT| 72.36 | 58.64 | 56.81 | 50.51 | 65.80 | 36.70 |
|Coder-Reviewer| - | 49.37 | 45.63 | 38.71 | 66.90 | 42.60 |
|Random| 59.88 | 45.20 | 26.68 | 32.55 | 37.06 | 22.78 |
|**SRank**|**75.31**| **59.99** | **60.55** | **53.99** | **69.66** | **43.07** |

**MBPP**
||WizardCoder34B|WizardCoder15B|CodeGen2.5-Instruct|StarCoder|Codex002|CodeGen16B|
|-----------|-----------|----------------|----------------|--|--|--|
|~~Incorrect Greedy~~|~~61.20~~|~~50.60~~|-|~~49.50~~|~~58.10~~|~~42.40~~|
|*Updated Greedy*|*60.42*|*51.29*|*42.86*|*45.90*|*58.10*|*42.40*|
|CodeT|63.39 | 58.18 | 55.02 | 58.05 | 67.70 | 49.50|
|Coder-Reviewer| - | 52.52 | 52.74 | 49.48 | 64.70 | 50.30|
|Random| 54.37 | 45.72 | 34.60 | 39.26 | 47.50 | 31.54 |
|**SRank**|**64.14**|**59.01**|**57.02**|**58.38**|**69.25**|**51.03**|

**References:**

1. Chen, B., Zhang, F., Nguyen, A., Zan, D., Lin, Z., Lou, J. G., & Chen, W. Codet: Code generation with generated tests. In ICLR. 2023*

---

### Author Response · Authors · 2023-11-21
**Overall Response**

Dear AC and Reviewers,

We would like to express our sincere gratitude to all the reviewers for their valuable feedback. We are greatly encouraged by the unanimous agreement among all the reviewers on the effectiveness of our proposed method. It is gratifying to note their acknowledge of the simplicity (**6Abb**, **BPa9**, **53rB**), innovation (**6Abb**) and potential applicability with a limited number of test cases (**BPa9**) of our method. In addition, we are pleased that they find our evaluation to be rigorous (**c5Kt**), convincing (**6Abb**), comprehensive (**53rB**) and appreciate our ablation study (**c5kt**, **6Abb**,**BPa9**).

In the following paragraphs, we address common questions and concerns raised by the reviewers.

1. **Concern on the process of generating and utilizing test cases from CodeLLM**

**Q3** [Reviewer **c5Kt**] “Is there any relation between what model generates a solution to the input problem and the test cases? If yes, does executing the test cases for the generated solution add value? Can we directly compare the test cases sampled from the model?”

**Q1** [Reviewer **6Abb**] "Why not just execute the test cases of the dataset instead of going to generate new test cases?"

**Answer:**

Regarding **Q3** [Reviewer **c5Kt**], we are not sure we understand your question correctly. We hope the reviewer can give some clarification regarding your questions? Here we give some information on the purpose behind generating test cases from language models in our reranking algorithm.

Similar to previous works [1, 2], we did not utilize the test cases available from datasets since it is challenging to obtain the available test cases for each piece of code in practice. Instead, we can prompt off-the-shelf LLMs to produce test cases, and we then utilize these test cases to choose the best-generated code candidates.

Generating test cases serves two purposes.

* First, the test inputs are used to execute the solutions. The execution outputs are then used for 1) clustering process by matching execution outputs of all solutions and 2) calculating the functional overlapping or interaction between clusters. Because the clustering process requires comparing execution outputs, the many and diverse test inputs are, the easier to distinguish the functionality of solutions. As a result, the more consistent the clusters will be. Moreover, increasing the number of test inputs also makes the overlapping computation more accurate. The benefit can be seen in Table 3. Performance improves when we increase the number of test cases, especially when the number of test cases is limited.

* Second, the generated test outputs are used as ground truth outputs to calculate the pass rate which is then used as a cluster feature, i.e vector V.

2. **Concern on limitation on benchmarking datasets**

**Q3** [Reviewer **6Abb**] "Therefore, it is suggested the authors validate SRank's effectiveness with more challenging datasets (e.g., Apps and CodeContests).”

**Q3** [Reviewer **BPa9**] “Benchmarks are limited. There are only two benchmarks, i.e., HumanEval and MBPP-S, and these benchmarks contain limited examples. We are not sure whether this method works or not in some large-scale datasets such as APPS.”

**Answer:**

We acknowledge the constraints associated with benchmark datasets. Given the temporal constraints during the rebuttal period, it was not feasible to extensively conduct experiments on more challenging benchmarks as highlighted by the reviewers, namely APPS and CodeContest. This is attributed to the considerable volume of problems, exemplified by the 5000 problems within APPS, and the substantial number of sampled sequences – 1000 solutions for each problem in the case of CodeContest. We affirm our commitment to undertake thorough experimentation and evaluation of our proposed method on these benchmarks, thereby providing the audience with a more comprehensive perspective in future iterations.

---

> ### Author Response · Authors · 2023-11-23
>
> 3. **Concern on the assumption that incorrect solutions are diverse and there is a low probability of functional overlap among incorrect solutions.**
>
> **Answer**:
>
> Our proposed method operates under the assumption that incorrect solutions are diverse and there is a low probability of functional overlap among incorrect solutions, so it is crucial to validate this assumption. Detailed results addressing this are presented in Section A of the Appendix. Quantitative findings indicate that the probability of lower functional overlap between incorrect solutions surpasses that of higher functional overlap, consistent with our assumption. In addition, we provide some qualitative examples through interaction matrices, revealing substantial functional overlap among correct solutions and low functional overlap among incorrect solutions.
>
> 4. **Reproducibility**
>
> **Answer:**
>
> Regarding reproducing our results, we provide the anonymous repository at https://anonymous.4open.science/r/SRank---ICLR-2024-BC32
>
>
> **References:**
>
> 1. Chen, B., Zhang, F., Nguyen, A., Zan, D., Lin, Z., Lou, J. G., & Chen, W. Codet: Code generation with generated tests. In ICLR. 2023
>
> 2. Ni, A., Iyer, S., Radev, D., Stoyanov, V., Yih, W. T., Wang, S., & Lin, X. V. Lever: Learning to verify language-to-code generation with execution. In ICML. 2023

---

### Meta-Review · Area_Chair_XkH2 · 2023-12-06

**Metareview:**

This paper proposes a method to rerank candidate code solutions from large language models by clustering those solutions based on their execution results on model-predicted test cases. The authors proposed to measure the functional overlap between different semantic clusters based on code execution results in order to identify the most representative cluster that “overlaps” with other clusters, hence the best code solution to the original problem. Experiments on a variety of code LLMs demonstrate improvements over existing reranking approaches, such as CodeT and Coder-Reviewer.

**Strengths**

* This paper is clearly presented and easy to follow (BPa9)

* The proposed approach is simple (6Abb, BPa9, 53rB), and the idea of modeling inter-cluster relationship sounds novel and interesting (6Abb).

* The method is also agnostic against the base LLMs used (53rB), and can be applied to realistic scenarios with a limited number of test cases (BPa9).

* The reranking method is effective on HumanEval/MBPP (c5Kt), with comprehensive ablations (c5Kt, 6Abb, 53rB)

**Weaknesses**

* Perhaps the most important issue is **practicality** —  the method requires 100 samples for simple code generation tasks like HumanEval/MBPP, making it questionable whether the approach can be applied to realistic settings where querying LLMs with 100x samples is costly and time consuming (c5Kt)

* In addition, evaluation is only conducted on simple code generation tasks, without evaluating on more challenging benchmarks such as CodeContests/APPS (6Abb, BPa9), which makes it difficult to assess if the proposed method would actually scale to those challenging scenarios where reranking is more necessary. There’s also no evidence if the method would be agnostic against different programming languages (c5Kt)

Given the above issues related to practicality and the limitation in evaluation datasets, the decision is "Reject".

**Justification For Why Not Higher Score:**

The above two weaknesses make it difficult to assess whether the proposed method could apply to real-world scenarios and more challenging code generation tasks. Therefore the current version does not quite reach the bar of a full paper.

**Justification For Why Not Lower Score:**

N/A

---

### Decision · Program_Chairs · 2024-01-16

Reject